

# Repair protocols for indirect monolithic restorations: a literature review

Lucas Saldanha da Rosa[1,*], Rafaela Oliveira Pilecco[1,*], Pablo Machado Soares[1], Marília Pivetta Rippe[1], Gabriel Kalil Rocha Pereira[1], Luiz Felipe Valandro[1], Cornelis Johannes Kleverlaan[2], Albert J. Feilzer[2] and João Paulo Mendes Tribst[2]

[1] Federal University of Santa Maria, Santa Maria, Rio Grande do Sul, Brazil
[2] Academic Centre for Dentistry Amsterdam (ACTA), Universiteit van Amsterdam en Vrije Universiteit, Amsterdam, the Netherlands
[*] These authors contributed equally to this work.

## ABSTRACT

Despite the advancements in indirect monolithic restorations, technical complications may occur during function. To overcome this issues, intraoral repair using resin composite is a practical and low-cost procedure, being able to increase the restoration's longevity. This review aimed to evaluate the need for repair and suggest a standardized repair protocol to the main indirect restorative materials. For this, studies were surveyed from PubMed with no language or date restriction, to investigate the scientific evidence of indirect monolithic restoration repair with direct resin composite. A classification to guide clinical decisions was made based on the FDI World Dental Federation criteria about defective indirect restorations considering esthetic and functional standards, along with the patient's view, to decide when polishing, repairing or replacing a defective restoration. Based on 38 surveyed studies, different resin composite intraoral repair protocols, that included mechanical and chemical aspects, were defined depending on the substrate considering resin-based, glass-ceramic or zirconia restorations. The presented criteria and protocols were developed to guide the clinician's decision-making process regarding defective indirect monolithic restorations, prolonging longevity and increasing clinical success.

## INTRODUCTION

All-ceramic and resin-based restorations have been widely indicated for oral rehabilitations, as they emerge as first options to computer-aided design computer–aided manufacturing (CAD-CAM). Added to this, CAD-CAM system stands out, since it is less time-consuming and reduces technical manufacturing variability (*Blatz & Conejo, 2019*). In addition to that, the advancement of ceramic and resin materials has facilitated the development of monolithic restorations (*Zhang & Kelly, 2017*). The use of these monolithic restorations has led to decreased chipping and delamination failures, which were previously associated with differences in the physical and mechanical properties of materials used in bilayer systems (*Swain, 2009*).

Corresponding author
João Paulo Mendes Tribst,
j.p.mendes.tribst@acta.nl

Dental ceramics can be classified according to their composition as either glass ceramics or polycrystalline ceramics (*Kelly, 2008*). Glass-ceramics are mainly composed of a silica matrix, which provides outstanding polish ability and aesthetic performance; and reinforcement by crystals such as leucite and lithium disilicate, responsible for the good mechanical behavior of these materials (*Zhang & Kelly, 2017*). As a polycrystalline ceramic composed solely of crystals arranged in various structures, zirconia stands out for its excellent mechanical properties (*Blatz, Vonderheide & Conejo, 2018*). Furthermore, it exhibits satisfactory aesthetic performance, particularly when compared to the use of metals. This is particularly true for the third generation of zirconia, which offers enhanced translucency compared to its first and second generations (*Stawarczyk et al., 2017*). These ceramic materials have a high modulus of elasticity, ranging from 84 to 210 GPa for lithium disilicate and zirconia, respectively, compared to the dentine structure (18 GPa). Zirconia ceramics have been modified over the years in order to have enhanced translucency and, recently, multilayered zirconias were introduced for high esthetic demanding regions besides advancements in sintering techniques (*Kongkiatkamon et al., 2023*). Ceramics are also brittle materials and susceptible to slow crack growth over time (*Zhang, Sailer & Lawn, 2013*). As an alternative, newer resin-based materials have been developed for the fabrication of restorations using CAD-CAM technology. According to manufacturers, the advantage of using resin-based materials lies in their elastic modulus, which is more similar to the tooth structure. Some authors state this can favor the stress distribution and, consequently, benefit the mechanical performance of the tooth-restoration assembly (*Dartora et al., 2019*; *Tribst et al., 2021a*), but it still debatable (*Costa et al., 2014*). Nonetheless, while in function in the oral cavity, resin-based materials also accumulate damage due to masticatory forces and hydrolytic degradation over time (*Saratti et al., 2021*).

Although all-ceramic restorations have shown satisfactory clinical performance (95% at 10 years—lithium disilicate, 91.3% at 10 years—zirconia, 90% at 10 years—feldspathic) (*Fasbinder, 2006*; *Valenti & Valenti, 2009*; *Sailer et al., 2018*), some maintenance procedures may be requested during their lifespan due to technical complications, as small chippings/fractures that have been reported (*Lemos et al., 2022*). In a survey taken in 2021 by the American Dental Association, with 400 clinicians, the three main reasons for deciding on a restoration repair were non-carious marginal defects, partial loss or fracture of restoration, and crown margin repair due to carious lesion (*da Costa et al., 2021*). To overcome this, intraoral repair kits are available on the market and are equally satisfactory for repairing resin composite and ceramic restorations, as reported by *Sanal & Kilinc (2020)*. When considering direct restorations, the success rate raises from 65.9% to 74.6% after 12 years when considering one repair as not a failure (*Casagrande et al., 2017*), and a similar benefit is probably extended to indirect restorations. Furthermore, repairs performed with resin-based composite in a single appointment by the direct technique are associated with advantages such as shorter treatment time and lower cost to the patient (*Loomans & Özcan, 2016*). Besides that, it propitiates a reduced need for re-interventions or additional tooth preparation, which can threaten pulp vitality and weak the tooth structure, allowing for more conservative and efficient restorative treatments (*Henry, 2009*; *Carrabba et al., 2017*; *FDI World Dental Federation, 2017*). However, some factors can influence the

behavior of the repaired restorative assembly, such as the type of surface treatment and patient conditions (*Casagrande et al., 2017*; *Kanzow et al., 2019*).

## To repair or not to repair

The discussion about the best repair protocol for each material and restoration type is not a new topic in the clinical and academic fields. However, it is important to note that before initiating any repairs, a thorough evaluation should be conducted to determine whether a repair is indeed indicated for the case at hand. Two perspectives can influence the decision-making process when noticing small fractures in the patient's restorations. The first is the remark by the clinicians themselves, that may notice even slight changes in restorations with the aid of potent operation lights, magnifying glasses, or intraoral cameras. The second is the patient's view, which will certainly notice medium and/or major alterations in the restoration. These two perspectives, combined, will determine the decision-making process on repairing, not repairing but making some adjustments like polishing or even the restoration replacement.

In 2010, the FDI World Dental Federation published clinical criteria for the evaluation of direct and indirect restorations—update and clinical examples (*Hickel et al., 2010*). The later document was intended to train and calibrate dental researchers, and improve clinical trial quality, student teaching and daily clinical practice. This document presents three major criteria with several subdivisions and different clinical conducts to be followed according to the obtained score. The first criteria are related to esthetic properties, the second to functional properties, and the third to biological properties. The esthetic properties evaluate surface gloss, staining, color match, translucency, and anatomical form. Functional properties evaluate fracture of material and retention, marginal adaptation, occlusal contour, wear, proximal anatomical form, radiographic examination (when applicable), and patient's view. The biological properties evaluate postoperative (hyper-) sensitivity and tooth vitality, recurrence of caries, erosion, abfraction; tooth integrity, periodontal response, adjacent mucosa; and oral and general health. All three categories were classified (with slight variations) as 1. Clinically excellent/very good; 2. Clinically good; 3. Clinically sufficient/satisfactory; 4. Clinically unsatisfactory (but reparable); 5. Clinically poor (replacement necessary). Based on this, one can meticulously evaluate the best clinical conduct for each restoration. Furthermore, it is encouraged to access the original material to observe the tables and photos in detail.

In the context of general dental practice, clinicians often encounter difficulties when attempting to comprehensively evaluate each restoration based on many criteria due to high patient flux on a daily basis. Added to this, *Heintze & Rousson (2010)*, three grades for evaluating chipping on restorations, as follow: Grade 1-small chipping receiving just some adjustments as polishing; Grade 2-moderate chipping that could be repaired with direct intraoral resin-based composite; Grade 3-severe chipping leading to the replacement of the entire prosthesis (*Heintze & Rousson, 2010*). These grades are directly related to the functional properties mentioned in the FDI criteria (*Hickel et al., 2010*). Considering this, it would be interesting to have an intermediate classification that could be easily applied

**Table 1** FDI simplified criteria for decision-making about defective indirect restorations.

|  | Polish | Repair | Replace |
|---|---|---|---|
| Esthetic | • Slightly dull surface with isolated pores<br>• Minor staining<br>• Color/translucency with minor deviations<br>• Form slightly deviated from normal | • Rough surface with voids<br>• Unacceptable staining<br>• Color/translucency with localized deviations<br>• Form unacceptable but repairable | • Very rough and/or plaque-retentive surface<br>• Severe staining not accessible for interventions<br>• Color/translucency unacceptable<br>• Form unsatisfactory/lost and not repairable |
| Functional | • Large hairline cracks and/or small chipping not involving marginal or proximal regions<br>• Small marginal fractures, minor irregularities<br>• Different wear rate than enamel<br>• Slightly strong contact<br>• Radiographically observed acceptable material excess, small step at the margin | • Moderate chipping in margin/proximal contacts<br>• Severe marginal fractures/irregularities<br>• Wear considerably exceeds normal enamel wearing<br>• Too weak contacts and possibly due to food impaction<br>• Radiographically observed material excess, but accessible and/or repairable steps | • Loss of restoration or multiple fractures<br>• Restoration with mobility, generalized major irregularities<br>• Excessive wear<br>• Too weak contacts, clear damage due to food impaction and/or pain/gingivitis<br>• Radiographically observed secondary caries, large gaps, apical pathology, restoration/tooth fracture |
| Patient's view | • Patient with minor criticism but no adverse clinical effects<br>• Some lack of chewing comfort | • Patient wants esthetical improvements<br>• Tongue irritation<br>• Desire for reshaping anatomic form | • Completely dissatisfied |

**Notes.**
Based on the FDI World Dental Federation: clinical criteria for the evaluation of direct and indirect restorations—update and clinical examples (*Hickel et al., 2010*).

daily and accurate enough to guide clinical decisions regarding repairing, or not, indirect restorations (Table 1).

Based on the aforementioned information, the objective of this review is to (1) summarize the scientific literature related to the protocols used for intraoral repairs of monolithic indirect restorations, also (2) to evaluate which criteria clinicians should consider for the need for repair and (3) suggest a standardized repair protocol specific to the ceramic/resin-based material used. The ultimate goal is to promote conservative restorative treatments that yield predictable outcomes and contribute to the longevity of the restorations.

## Survey methodology

The studies were surveyed from PubMed with no language or date restriction, to investigate the scientific evidence of indirect monolithic restoration repair with direct resin-based composite. The adopted search strategy was based on the main materials and some of the most used brands: (("glass ceramics" [All Fields] OR "zirconia"[All Fields] OR "indirect resin composite"[All Fields] OR "lava ultimate"[All Fields] OR "tetric cad"[All Fields] OR "lithium disilicate"[All Fields] OR "cerasmart"[All Fields] OR "feldspathic"[All Fields] OR "5y-psz"[All Fields] OR "4y-tzp"[All Fields] OR "monolithic" [All Fields]) AND "repair"[All Fields]). There were included studies that evaluated direct resin-based composite repairs to monolithic indirect ceramic or composite materials. Based on 278 surveyed entries and references of relevant studies on the field, thirty-eight studies were included and qualitatively analyzed. Different resin-based composite intraoral repair

protocols, which included mechanical and chemical aspects, were defined depending on the substrate considering resin-based, glass-ceramic, or zirconia restorations.

# RESULTS

## Resin-based materials

Intraoral repairs of CAD-CAM resin-based materials, such as Tetric CAD (Ivoclar AG, Schaan, Liechtenstein), Lava Ultimate (3M ESPE, St. Paul, MN, USA), Cerasmart (GC), Ambarino High-Class (Creamed, Apulia, Italy), Paradigm MZ100 (3M ESPE, St. Paul, MN, USA), Shofu Block HC (Shofu, San Marcos, CA, USA), Brilliant Crios (Coltene, Switzerland), and Grandio Blocs (Voco, Cuxhaven, Germany), are slightly less complex than ceramics due to the compatibility between materials of the same nature (substrate to be repaired and repair material) (*Bayraktar, Arslan & Demirtag, 2021*). One manufacturer (Ivoclar AG, Schaan, Liechtenstein) mentions the preparation of the substrate before the repair through sandblasting using, for example, aluminum oxide ($Al_2O_3$) particles (Tetric CAD, Scientific Documentation). Air-abrasion for dental application can be performed mainly using $Al_2O_3$ or tribochemical silica airborne-particles, with different steps and grain-sizes. Considering the surface treatment of resin-based materials before repair, the literature seems to be consistent with the advantages of air-abrasion compared to other protocols. Some studies have reported a superior bond strength after air-abrasion, regardless of which particle was used, compared to etching (*Tatar & Ural, 2018*; *Sismanoglu et al., 2020*; *Veríssimo et al., 2020*; *Şişmanoğlu et al., 2020*) and grinding protocols (*Stawarczyk, Krawczuk & Ilie, 2015*; *Wiegand et al., 2015*). However, other studies reported a similar behavior compared to diamond bur grinding (*Wiegand et al., 2015*; *Güngör et al., 2016*; *Arkoy & Ulusoy, 2022*). Considering the different particles used for air-abrasion, previous authors reported a superior performance for the tribochemical silica airborne-particles (*Sismanoglu et al., 2020*; *Sismanoglu et al., 2020*; *Bayazıt, 2021*), while others reported a similar adhesive outcome compared to traditional $Al_2O_3$ (*Wiegand et al., 2015*; *Subaşı & Alp, 2017*; *Loomans et al., 2017*; *Arpa et al., 2019*; *Moura et al., 2020*; *Sismanoglu et al., 2020*).

As mentioned before, hydrofluoric acid (HF) etching was also reported as a possible surface treatment (*Güngör et al., 2016*; *Loomans et al., 2017*; *Gul & Altınok Uygun, 2020*; *Bayazıt, 2021*; *Bayraktar, Arslan & Demirtag, 2021*). However, there are considerable differences (of etching time and concentration) in the protocols of application (Table S1), making it difficult to define the advantage of such surface treatment. In addition, considering the possible hazardous effect of HF intraoral use on the dental tissues and on the resin-based composite material itself (*i.e.,* water penetration and consequent disorganization of the siloxane network), the possible adequate result does not substantiate its use for surface treatment before resin repair (*Özcan, Allahbeickaraghi & Dündar, 2012*). In a recent systematic review, the air-abrasion with $Al_2O_3$ or grinding with diamond bur resulted in a higher bond strength compared to the tribochemical silica airborne-particles system and hydrofluoric acid for Lava Ultimate resin composite (*Moura et al., 2022*).

Other treatments can involve the use of specific repair systems, such as GC Repair (GC), Cimara System (VOCO, Cuxhaven, Germany), Porcelain Repair (Ultradent, South

Jordan, UT, USA), Clearfil Repair System (Kuraray, Okayama, Japan), Z-Prime Plus (Bisco, Anaheim, CA, USA), and Ceramic Repair (Ivoclar AG, Schaan, Liechtenstein), which also seems to be a promising option evolving an etching step (HF or phosphoric acid) and subsequently primers and/or adhesives (*Üstün, Büyükhatipoğlu & Seçilmiş, 2018*; *Gul & Altınok Uygun, 2020*). The use of lasers, such as neodymium-doped yttrium aluminum garnet (Nd: YAG), erbium-doped yttrium aluminum garnet (Er: YAG), erbium, chromium-doped yttrium, scandium, gallium, and garnet (ErCr: YSGG) laser, has also been reported, but with contradictory results compared to more traditional approaches (*Bahadır & Bayraktar, 2020*; *Bayraktar, Arslan & Demirtag, 2021*; *Arkoy & Ulusoy, 2022*).

Another clinical step is associated with the application of an adhesive layer at the adhesive interface, applied with the aim of facilitating the bonding between the materials by increasing the surface wettability and promoting a sufficient bond strength *per se* through the bond with unpolymerized resin monomers (*Arkoy & Ulusoy, 2022*; *Arpa et al., 2019*; *Bayazıt, 2021*; *Loomans et al., 2017*; *Sismanoglu et al., 2020*; *Veríssimo et al., 2020*). The adhesive layer effect can be enhanced when a previous grounded surface was used as surface treatment (*Arpa et al., 2019*; *Bahadır & Bayraktar, 2020*; *Veríssimo et al., 2020*; *Bayraktar, Arslan & Demirtag, 2021*). Otherwise, the application of a silane coupling agent has been reported mixed into the repair protocols (Table S1). In dental applications, silanes are used to enhance the adhesion between dissimilar materials, particularly when bonding composite materials to inorganic substrates like ceramics, glass, or metal. For the repair of composite materials, silane could be used to bond the ceramic fillers that are exposed on the restoration's surface. Only a few studies reported a beneficial effect of the application of silane (*Wiegand et al., 2015*; *Güngör et al., 2016*; *Sismanoglu et al., 2020*), based on the interaction between the silane and the resin matrix and/or the adhesive layer (*Matinlinna, Lung & Tsoi, 2018*). Meanwhile, others reported no difference (*Loomans et al., 2017*; *Tatar & Ural, 2018*; *Arpa et al., 2019*). It is worth mentioning that there are important methodological differences among the primary studies and, considering that the effect of silane has been reported as material and surface treatment-dependent (*Loomans et al., 2017*), studies considering this factor isolated are necessary.

In this sense, it is also important to highlight that different resin-based materials may respond differently to the surface treatments and to the repair itself, considering the differences in composition and mechanical properties (*Wiegand et al., 2015*; *Güngör et al., 2016*; *Loomans et al., 2017*; *Demirel & Baltacı oğlu, 2019*; *Gul & Altınok Uygun, 2020*; *Sismanoglu et al., 2020*; *Moura et al., 2022*; *Arkoy & Ulusoy, 2022*). Also, some newly launched resin-based materials are not robustly investigated by primary studies, remaining a scientific gap regarding their properties. In this sense, another factor is the adhesive systems used, which also according to the composition and the reaction with the substrate material, can perform differently (*Stawarczyk, Krawczuk & Ilie, 2015*; *Demirel & Baltacı oğlu, 2019*; *Arpa et al., 2019*), especially when considering long-term behavior. To assess that, thermocycling is frequently used to simulate aged composite before repairing (Table S1), even though, most studies used a relatively low number of cycles (5,000 thermal cycles) (*Armstrong et al., 2017*). Despite that, some studies reported that thermocycling

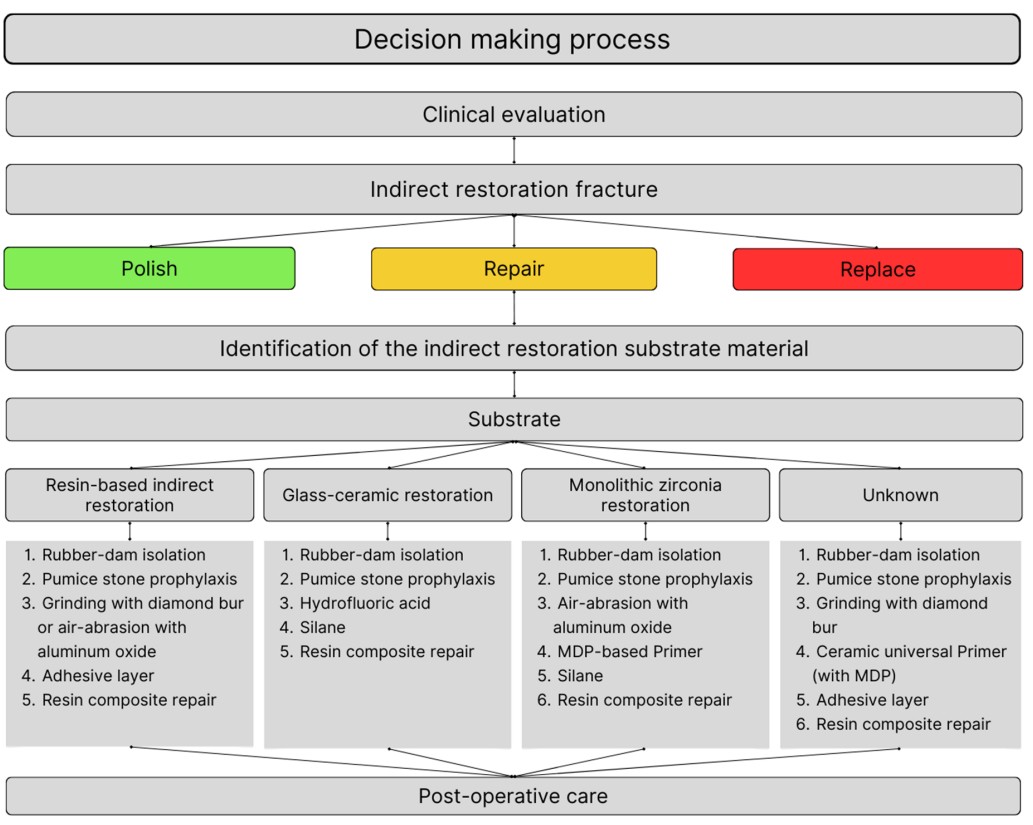

**Figure 1** Flowchart outlining the clinical decision-making process for repairing indirect monolithic restorations based on the substrate.

affects negatively the repaired interface, inducing a worst bond strength compared to the non-aged condition (*Güngör et al., 2016*; *Subaşı& Alp, 2017*; *Loomans et al., 2017*).

In conclusion, it seems that the air-abrasion with $Al_2O_3$ particles or grinding with a diamond bur, followed by the application of an adhesive system can be sufficient to promote a reliable and strong adhesion between a resin-based composite substrate and a resin composite repair (Fig. 1). However, there is still a lack of consensus regarding the effect of different surface treatments and silane application on the adhesive and mechanical performance considering more clinically realistic and complex situations.

## Glass-ceramic materials

Unlike resin-based materials, repairing glass-ceramic materials is far more challenging since the nature of the repairing material and the restoration to be repaired have fewer points in common. Glass-ceramics are known for its high-volume glassy matrix (based on silica) and this class of material can vary in reinforcement particles, which will determine whether the ceramic will be more aesthetic or mechanically resistant (*Kelly & Benetti, 2011*).

The traditional protocols for bonding glass-ceramics are performed extraorally by the laboratory technician or by the dentist itself. In this sense, it is commonly used surface conditioning materials that are aggressive chemical solutions with high toxicity and

may cause severe acid tissue burns and can even be forbidden by dentists to have it on clinics in many countries (*Höller et al., 2022*). The main etchant used in glass-ceramics is 5–10%HF acid usually applied for times between 20 s and 60 s depending on the etchant concentration and glass content in the material (*Riesgo et al., 2023*). However, the HF use can be harmful to the soft tissues; and when applied in a repair procedure, the patient should be appropriately protected with rubber-dam isolation. Furthermore, this type of protection is associated to improved bond strength, especially to enamel, being mandatory to any case involving bonding to this dental structure (*Falacho et al., 2023*). This etchant is classically followed by the application of a silane-containing substance, hence silane, a bi-functional molecule that can link the glass ceramic's inorganic portion to the resin cement's organic matrix (*Matinlinna, Lung & Tsoi, 2018*). In the case of direct repair, the silane molecule would link not to a resin cement, but to an adhesive system used prior to resin-based composite restoration. Its use is the most explored repair method in the literature, and tends to be the most successful when bonding glass-ceramic materials to direct resin composites (Table S1). The other commonly reported repair methods are air-abrasion with $Al_2O_3$ and tribochemical silica airborne-particles systems. The latest one, due to its silica deposition by impact on the ceramic surface, is more effectively bonded to silane molecules and should make a more favorable bonding to the repair materials (*Al-Thagafi, Al-Zordk & Saker, 2016*). Despite the similar indications, the literature reports that HF etching promotes better adhesion to direct resin composites than these systems for repair surface treatment (*Colares et al., 2013*; *Duzyol et al., 2016*; *Ataol & Ergun, 2018*; *Veríssimo et al., 2020*; *AL-Turki et al., 2020*; *Kilinc, Sanal & Turgut, 2020*; *Höller et al., 2022*; *Aladağ & Ayaz, 2023*). In one study, when combining tribochemical silica airborne air-abrasion with HF etching, the results seemed favorable, however, this was probably promoted by the HF etching itself (*Huang, Wang & Gao, 2013*). In a few studies, these systems seem equivalent to more established methods, like HF etching (*Atala & Yeğin, 2022*; *Erdemir et al., 2014*; *Maawadh et al., 2020*) and rarely appeared as concrete alternatives (*Al-Thagafi, Al-Zordk & Saker, 2016*).

The most practical alternative for the clinician is surface grinding with a diamond bur. This method, however, seems not to be a solid alternative for glass-ceramics, since the literature shows that it provides poor adhesion to the repair resins when compared to other methods, even when a silane-containing substance is used before the restoration (*Huang, Wang & Gao, 2013*; *Erdemir et al., 2014*; *Duzyol et al., 2016*; *Veríssimo et al., 2020*; *Bayraktar, Arslan & Demirtag, 2021*). By the nature of this method, it promotes macro-irregularities, with secondary and minor micromechanical interlocking when compared to other methods, and this may be a reason for short-term failure (*Erdemir et al., 2014*).

For direct repair of monolithic dental restorations, the use of a non-toxic product seems like an essential consideration to be taken. Different substances emerged as surface treatment options to traditional approaches, such as the application of a self-etching ceramic primer, like Monobond Etch & Prime (Ivoclar AG, Schaan, Liechtenstein) (*Dapieve et al., 2020*; *Tribst et al., 2021b*). This primer is sufficiently acidic to promote surface alterations to create micromechanical retentions and has the incorporation of silane molecules, being able to establish siloxane bonds between the resin material and the ceramic material

(*Matinlinna, Lung & Tsoi, 2018*; *Tribst et al., 2018*). Just a few studies investigated this self-etching primer (*Ueda et al., 2021*; *Höller et al., 2022*) as a surface treatment alternative for direct repair of glass-ceramic restorations. These studies did not compare this primer to HF etching, but to phosphoric acid etching (*Ueda et al., 2021*) or air-abrasion (*Höller et al., 2022*), and they indicated that Monobond Etch & Prime presented high stability over time and a promising strategy for glass-ceramic's direct repair. More research employing this self-etching primer as a repair surface treatment should be conducted, especially comparing it to the classical HF etching.

Beside using air abrasion, mechanical or chemical treatment; the ceramic surface can also be altered by light. However, when low-level laser therapy on the ceramic surface, such as Nd: YAG, Er: YAG, Er,Cr: YSGG, and methylene blue photosensitizer (MBPS) were used, other treatments excelled (*Erdemir et al., 2014*; *Ebrahimi Chaharom et al., 2018*; *Ataol & Ergun, 2018*; *Maawadh et al., 2020*; *Kilinc, Sanal & Turgut, 2020*; *Bayraktar, Arslan & Demirtag, 2021*). The surface pattern generated by laser seems to be smoother compared to air-abrasion or HF etching, and this may impair the microinterlocking between the ceramic and the direct resin composite (*Erdemir et al., 2014*; *Ataol & Ergun, 2018*). Besides showing some benefit for luting glass-ceramics (*Feitosa et al., 2021*), the use of low-level laser therapy does not appear to be a relevant surface conditioning method in scenarios of glass ceramics direct repairs.

Based on the previous literature, lithium disilicate glass-ceramic emerges as the most extensively researched glass-ceramic among the available options. For this ceramic, HF surface etching and silane application was the most successful method (*Huang, Wang & Gao, 2013*; *Duzyol et al., 2016*; *Ebrahimi Chaharom et al., 2018*). Other recurrent cited ceramics include feldspathic and zirconia-reinforced lithium silicate. The studies regarding these two materials do not point to a preferred method for zirconia-reinforced glass ceramic (ZLS), while HF seems to be the best approach for feldspathic ceramic (*AL-Turki et al., 2020*; *Bayraktar, Arslan & Demirtag, 2021*).

The majority of glass ceramics repair studies use an adhesive system prior to the resin composite application. Although, there is no scientific base for using adhesive systems in ceramics, and this step may impair the restoration mechanical properties (*Velho et al., 2022*). For this reason, more studies are needed exploring such thematic in the repair context. In this sense, the recommended surface treatment option for direct repair of glass ceramics is HF etching with posterior silane application (Fig. 1).

## Polycrystalline ceramic materials

Differently from glass-ceramics, the arrangement of yttria-stabilized zirconia (YSZ) makes it even more challenging to promote adequate adhesion with resin composite (*Zarone et al., 2019*). Since YSZ does not exhibit sensitivity to hydrofluoric acid etching (*Smielak & Klimek, 2015*), different surface treatments have been suggested to overcome these bonding limitations (*Ozcan, Nijhuis & Valandro, 2008*; *Sakrana & Özcan, 2017*). However, there is a lack of studies evaluating the performance of repair procedures when considering the latest generation of zirconia, which presents a higher amount of cubic phase and percentage of

yttrium stabilizer, making it more translucent and indicated for monolithic restorations in both anterior and posterior regions (*Stawarczyk et al., 2017*).

Four studies that evaluated repair procedures for monolithic restorations made of translucent zirconia were found and have been discussed in the present review (*Klaisiri et al., 2022*; *Ordueri, Ateş & Özcan, 2023*; *Greuling et al., 2023*; *Aladağ & Ayaz, 2023*). Three of these studies evaluated the shear bond strength between YSZ and the repair resin composite (*Klaisiri et al., 2022*; *Ordueri, Ateş & Özcan, 2023*; *Aladağ & Ayaz, 2023*), while only one evaluated the fracture resistance of repaired crowns when compared to the non-repaired ones (*Greuling et al., 2023*). The results of this last study showed that repaired crowns after endodontic treatment through the use of universal adhesive presented lower fracture resistance than the group with no treatment, regardless of the presence of thermocycling. This may be explained by the introduction of defects and microcracks in the translucent zirconia during the trepanation for endodontic purposes, which may concentrate stresses during the mechanical load application leading to reduced strength. Besides, the Young's modulus between the present adhesive system and resin composite are too different from YSZ's (*Ivanoff, Hottel & Garcia-Godoy, 2018*; *Soares et al., 2021*), thus generating a different stress distribution which affects the mechanical performance. Even so, the fracture resistance found for the repaired group with universal adhesive was considered strong enough for clinical use (*Greuling et al., 2023*).

Different surface treatments were reported in the mentioned studies, being the use of air-abrasion protocols with $Al_2O_3$ (*Klaisiri et al., 2022*; *Ordueri, Ateş & Özcan, 2023*; *Aladağ & Ayaz, 2023*), with or without silica coating, the most used approach. This is in accordance with previous studies considering dental zirconia (*Ozcan, Nijhuis & Valandro, 2008*; *Sakrana & Özcan, 2017*; *Altan, Cinar & Tuncelli, 2019*), where the air-abrasion was also the most widely used approach to modify the surface topography and roughness of the polycrystalline material, and by that to increase the bond strength with resin composites for direct repair. *Aladağ & Ayaz (2023)* compared this approach to other surface treatments such as laser treatment, HF etching, a combination of both, or even only the use of adhesive systems (chemical bonding). The air-abrasion with 50 μm $Al_2O_3$ particles was able to increase the repair bond strength for YSZ but showed intermediate values of bonding when compared to the other groups (Table S1). The HF treatment was similar to the absence of mechanical treatment (only adhesive), and both presented lower bond strength between YSZ and repair resin composite when compared to the laser therapy, which presented the highest values. These findings are corroborated by the literature, which reports that HF etching does not affect the zirconia surface (*Zarone et al., 2019*). Laser irradiation has been considered as an alternative surface treatment for zirconia, since it may modify the ceramic surface and create irregularities to increase the mechanical interlocking to resin-based materials (*Usumez et al., 2013*). However, only one study was found considering such treatment for the repair of translucid zirconia monolithic restorations. Therefore, it is encouraged to conduct further studies on this topic. Besides the mechanical surface modifications, chemical approaches such as the use of 10-methacryloyloxydecyl dihydrogen phosphate (MDP)-containing primers are also essential to increase the bonding affinity of YSZ to the resin matrix, by the increase of

the ceramic wettability and chemical affinity by the phosphate monomer (*Kitayama et al., 2010*). All studies reported the use of primers and/or adhesive systems after the surface treatment (*Klaisiri et al., 2022*; *Ordueri, Ateş & Özcan, 2023*; *Greuling et al., 2023*; *Aladağ & Ayaz, 2023*), or even a universal adhesive only (*Greuling et al., 2023*). Just one study compared the combination of mechanical and chemical approaches to only chemical and reported that higher values of bond strength were found when a mechanical treatment was performed before the primer and adhesive application (*Aladağ & Ayaz, 2023*). Also, one study evaluated if the number of applications of a phosphate-containing primer prior to the use of an adhesive affected the bond strength of translucent YSZ to a resin composite, and reported that three applications resulted in higher bond strength valuer when compared to less or no application (only adhesive), by the increase of the phosphate monomer concentration (*Klaisiri et al., 2022*). However, no benefit was observed when applying more than three times, hence the saturation of MDP must be avoided.

The concept of utilizing an adhesive system when fixing ceramic restorations, as observed in glass-ceramic studies, has also been applied to zirconia research. As previously stated, it is not recommended to use adhesive systems in repairing polycrystalline restorations until studies confirm the necessity of such a step as a standard protocol. Thus, it seems that a protocol that includes air-abrasion followed by the use of an MDP-containing primer may be indicated for repair procedures (Fig. 1). However, more studies evaluating these approaches are essential when considering translucent zirconia for monolithic restorations.

## DISCUSSION

### Direct or indirect repair—alternative techniques

In cases where screw-retained implant-supported restorations or removable prostheses require repair, indirect restoration can be done in the prosthetic laboratory as an alternative to direct intraoral repair. In the case that the prosthetic piece can be removed, and a porcelain or ceramic fragment can be fabricated onto the chipped area by the lab. When the restoration cannot be removed, an impression of the chipped area can be performed and the new fragment can be bonded intraorally (*Proaño et al., 2021*). Even so that repairing using bonded ceramic fragments indirectly may result in a higher load-to-fracture than with direct restoration with resin composite, it is required a laboratory procedure, which takes more time and has a higher cost to the patient (*Kumchai et al., 2020*). Also, in the first scenario, the restoration needs to be put in the oven again, which may induce crack formation and the worst flexural strength of an already damaged material (*Gonuldas, Yılmaz & Ozturk, 2014*; *Subaşıet al., 2022*).

In this sense, an alternative technique has been described for screw-retained implant-supported restorations, in which the restoration is repaired extraorally with a direct resin composite (*Proaño et al., 2021*). After the proper surface treatment (according to the restorative material) (Fig. 1), a direct resin composite is bonded to the fractured surface. On one hand, it is possible to perform the repair at the same appointment with an easy and affordable treatment. On the other hand, as the color stability of resin composite performs differently than dental ceramics, as time passes it may need an esthetic repair

(*Lu et al., 2005*; *Paolone et al., 2023*). In those cases, for both implant- or tooth-supported restorations, the clinician can choose to fabricate an indirect dental ceramic fragment and to bond into the preexisting ceramic restoration without removing it, therefore with a minimally invasive approach (*Strasding et al., 2018*).

## Clinical considerations

When (*Kanzow et al., 2017*) asked 1,805 German dentists about the used procedures to repair ceramic restorations, after surface cleaning (done by 71.0%) and surface roughening (done by 70.1%), HF use was reported as a common repair surface conditioning procedure by 43.0% of the participants. The silane application was reported by 60.7%, while the use of adhesive systems was reported by 85.6% of the dentists. The procedures used were slightly different for composite restorations, with almost the same rate performing surface cleaning protocols (69.9%) and surface roughening with diamond bur (75.5%). The most distinct difference relies on the phosphoric acid application before the repair procedure, reported by 67.3% of the respondents. Interestingly, air abrasion, which is a highly efficient surface treatment for composite repair was reported only by 15.1% of the dentists asked.

The attempt in extending the restorations' longevity using repair procedures starts in the first contact with the failed restoration. For all materials, surface cleaning must be carried out for removing plaque and other contaminants, especially using an abrasive substance as in pumice stone prophylaxis (*Barchetta et al., 2021*). Another key point is moisture control. The oral environment has a large presence of fluids that can jeopardize the adhesion mechanisms (*Tsujimoto et al., 2018*) and for this reason, the storage protocol for aging in *vitro* studies is in 37 °C water. With this in mind, proper tooth isolation with a rubber-dam must be used in order to improve adhesion in repair scenarios (*Höller et al., 2022*). However, besides using the best approaches to extend the restorations' service time, it is known that repairs in endodontically treated teeth and patients that use removable dentures show reduced longevity for repaired restorations (*Casagrande et al., 2017*).

Questions regarding the best resin composite for repair procedures were also addressed, however, there is no consensus about the theme (*Höller et al., 2022*). The resin composites used for direct repairs of indirect restorations vary in the filler's particle size. From the studies recovered from the literature, for indirect resin composite repair, nanohybrid-filled direct resin composites were the most used, followed by nano-filled, microhybrid, and flowable resin composites. In the same manner, the nanohybrid resin composites stood out, followed by microhybrid, nanoceramic, and one study with a supra-nano resin composite (*Aladağ & Ayaz, 2023*) for glass ceramic repair. The variation in the filler particle size of resin composites seemed to not influence the observed outcomes and there is no alteration in the repair protocols according to the material composition. Another class of resin composite used was the self-adhesive resin composites in a few studies (*Erdemir et al., 2014*; *Karci et al., 2018*; *Sanal & Kilinc, 2020*). They point to the fact that HF etching or tribochemical silica airborne-particles air-abrasion followed by silane are still solid options for surface treatment when using these composites. Besides, their findings indicate that these materials' bond strength to the substrate is not superior to the traditional used resin composites with conventional direct repair approaches for glass ceramics.

## Post-operative care

After the repair with resin composite application, a finishing and polishing protocol must also be considered to enhance the restoration longevity. Moreover, technical complications have been reported for indirect restorations, such as small chipping, wear, and fractures (*Valenti & Valenti, 2009*; *Sailer et al., 2018*; *Lemos et al., 2022*). Among the reasons for these failures, unbalanced occlusion, and parafunctional habits such as bruxism are usually reported (*Lemos et al., 2022*). Thus, a minacious occlusal evaluation and finishing procedures must be performed by clinicians after the repair protocol (*Yap, Ang & Chong, 1998*). Besides, staining, biofilm accumulation, and secondary caries can occur on rough surfaces. In this context, the use of low abrasive instruments is recommended to obtain a regular and polished repaired surface, and the use of decreasing abrasive discs has been reported for such procedures when considering a repair scenario (*Shafiei, Berahman & Niazi, 2016*; *Greuling et al., 2023*).

Another point of discussion is about the time to make the repair finishing. A previous study evaluated the effect of performing finishing at different times after the repair process on the microleakage at the interface (*Shafiei, Berahman & Niazi, 2016*). The repaired resin composite restorations were finished with abrasive discs three times: immediately, after 20 min; and after 24 h. The microleakage was significantly higher in the group that was immediately finished. However, there was no difference between the groups finished after 20 min and 24 h. Thus, the repair protocol and finishing/polishing of the restoration can be made in a single section without impairing the interface sealing, besides assuring a polished surface and satisfactory occlusion. To the author's knowledge, there are no studies evaluating such factors when considering the repair of ceramic materials, so future studies on this sense are encouraged.

## CONCLUSION

When a patient presents an indirect monolithic restoration that appears to be chipped or fractured, an extensive and detailed clinical evaluation must be performed. First, the dentist needs to understand the reason for the restoration failure, for preventing it from happening again and prolong the longevity of the rehabilitation. Second, the restoration material needs to be identified for the application of the proper surface treatment and clinical steps, as previously described. The adequate use of the medical file is crucial and in cases where this information is missing or cannot be assured, there is no universal protocol standardized and based on scientific evidence for repairing the restoration. With that in mind, the dentist can indicate the full replacement of the restoration or try to perform a clinical protocol to conserve the restoration for a little longer, depending on how extensive the fracture/chipping is. For the latter, different protocols have been herein described and should be used as reference. In this sense, efforts should be made to properly report and identify the materials considering the whole clinical scenario.

In light of all the evidence, the present review summarized the scientific literature regarding protocols used for intraoral repairs of monolithic indirect restorations and observed that repairing with resin composite seems to be an interesting alternative for

prolonging the longevity of indirect monolithic restorations. Simplified criteria for decision making in the decision for repairing failed restorations was proposed and a standardized repair protocol specific to the ceramic/resin-based material used was elaborated to guide clinicians based on scientific evidences aiming clinical success.

### Funding

This work was supported by the abroad visiting-researcher scholarship #201081/2022-9 by the Brazilian National Council for Scientific and Technological Development—CNPq to L.S.R. Doctorate's scholarship by the Brazilian Federal Agency for Coordination of Improvement of Higher Education Personnel—CAPES (Finance code 001); a doctorate scholarship, #140118/2022-5, and abroad visiting-researcher scholarship, #201080/2022-2, by the Brazilian National Council for Scientific and Technological Development—CNPq to R.O.P.; an abroad visiting-researcher scholarship #888877.17140/2022-00 at CAPES/PrInt Program, Smart Materials Project to P.S.M. Doctorate's scholarship by the Brazilian Federal Agency for Coordination of Improvement of Higher Education Personnel—CAPES (Finance code 001). The funders had no role in study design, data collection and analysis, decision to publish, or preparation of the manuscript.

### Grant Disclosures

The following grant information was disclosed by the authors:
The Brazilian National Council for Scientific and Technological Development—CNPq: # 201081/2022-9.
The Brazilian Federal Agency for Coordination of Improvement of Higher Education Personnel—CAPES (Finance code 001). - R.O.P.: doctorate scholarship: #140118/2022-5, #201080/2022-2.
the Brazilian National Council for Scientific and Technological Development—CNPq. - P.S.M abroad visiting-researcher scholarship: #888877.17140/2022-00.
The Brazilian Federal Agency for Coordination of Improvement of Higher Education Personnel—CAPES (Finance code 001).

### Competing Interests

João Paulo Mendes Tribst is an Academic Editor for PeerJ.

### Author Contributions

- Lucas Saldanha da Rosa conceived and designed the experiments, performed the experiments, analyzed the data, prepared figures and/or tables, and approved the final draft.
- Rafaela Oliveira Pilecco conceived and designed the experiments, performed the experiments, analyzed the data, prepared figures and/or tables, and approved the final draft.

- Pablo Machado Soares conceived and designed the experiments, performed the experiments, analyzed the data, prepared figures and/or tables, and approved the final draft.
- Marília Pivetta Rippe conceived and designed the experiments, analyzed the data, authored or reviewed drafts of the article, and approved the final draft.
- Gabriel Kalil Rocha Pereira conceived and designed the experiments, analyzed the data, authored or reviewed drafts of the article, and approved the final draft.
- Luiz Felipe Valandro conceived and designed the experiments, analyzed the data, authored or reviewed drafts of the article, and approved the final draft.
- Cornelis Johannes Kleverlaan conceived and designed the experiments, analyzed the data, authored or reviewed drafts of the article, and approved the final draft.
- Albert J Feilzer conceived and designed the experiments, analyzed the data, authored or reviewed drafts of the article, and approved the final draft.
- João Paulo Mendes Tribst conceived and designed the experiments, analyzed the data, authored or reviewed drafts of the article, and approved the final draft.

## Data Availability

The raw reference list (BibTeX) is available in the Supplementary File 1. The raw data shows all manuscripts used in the present review.

## Supplemental Information

Supplemental information for this article can be found online at http://dx.doi.org/10.7717/peerj.16942#supplemental-information.

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
