# Peer review of "Repair protocols for indirect monolithic restorations: a literature review"

_PeerJ, doi:10.7717/peerj.16942_

## Round 0.1 · original submission · Major Revisions

Please address to the reviewer comments and resubmit the revision.

Reviewer 1 has suggested that you cite specific references. You are welcome to add it/them if you believe they are relevant. However, you are not required to include these citations, and if you do not include them, this will not influence my decision.

Reviewer 1 ·

Basic reporting

This article gave criteria and protocols to guide the clinician's decision-making process regarding defective indirect monolithic restorations and these are useful clinically. Some comments.

There are recent improvements in the sintering protocols. In the Introduction, it will be informative to add various types of Zirconia in Dentistry and various sintering protocols. https://www.ncbi.nlm.nih.gov/pmc/articles/PMC10351515/

The categories. Line 151-155. It is better to add in Tabular or diagrammatic for the reader for better visualization.

Line 190-200. Add References.

There seems to be some confusion. The authors also added composite resin repair. Please 523-524. Please separate or remove these from the article as the article focuses on monolithic ceramic materials.

Please add references to Table 1.

The article needs editing and simplification.

Experimental design

The study design is good.

Validity of the findings

no comment

Reviewer 2 ·

Basic reporting

everything seems ok

Experimental design

everything seems ok

Validity of the findings

everything seems ok

Additional comments

It is an interesting review, well designed, good referenced and with good clinical suggestions

Annotated reviews are not available for download in order to protect the identity of reviewers who chose to remain anonymous.

Reviewer 3 ·

Basic reporting

The article is written in an understandable language.
Background is sufficient and references are appropriate and sufficient in number.
The tables and figures used in the article are also appropriate

Experimental design

Article content is within the Aims and Scope of the journal.
Accessing more data by scanning areas other than PubMed will further increase the impact of the study. Some articles can be accessed by scanning outside of PubMed on this subject.

Validity of the findings

Discussion and conclusion sections are handled comparatively in line with the data obtained.

Additional comments

This article, written as a review, is designed to shed light on researchers and clinicians and is written in an understandable language.
Accessing more data by scanning areas other than PubMed will further increase the impact of the study. Some articles can be accessed by scanning outside of PubMed on this subject.

Reviewer 4 ·

Basic reporting

The authors’ stated aim was to investigate the scientific evidence of indirect monolithic restoration repair with direct resin composite. Whilst this is an interesting and timely topic in the field of restorative dentistry it is regratable that the authors have not achieved their principal aim. This is because this manuscript is based on a narrative report. To meet their aim in terms of investigating the scientific evidence on the topic the authors should have performed a systematic review.

The authors’ stated aim was to investigate the scientific evidence of indirect monolithic restoration repair with direct resin composite. Whilst this is an interesting and timely topic in the field of restorative dentistry it is regrettable that the authors have not achieved their principal aim. This is because this manuscript is based on a narrative report. To meet their aim in terms of investigating the scientific evidence on the topic the authors should have performed a systematic review.

The authors are referring to their narrative review as a "critical review". However, for it to be critical the authors should not have merely presented findings and conclusions from other studies, but rather should have critically appraised their cited papers, thereby also discussing the bias and shortfalls in terms of study design and/or reporting in some of the papers the authors are citing. Unfortunately, this omission has a detriment effect on the scientific merit and external validity of the presented manuscript.

The literature references do not include many key papers on the topic indirect restorations repair - too many to mention in this review. In particular, there are several not included studies that looked at intraoral repair kits.

Additional points of note include:
(1) the introduction being too lengthy and too extensive on material properties, i.e., lines 95-101.
(2) in the section on 'To repair or not to repair' a third and most important perspective that can influence clinical decision-making when noticing small fractures has not been mentioned: identifying and eliminating possible causative factors. Although this is mentioned by the authors in their conclusion, this section would have been the ideal section to mention this important perspective.
(3) The manuscript reads like a presentation of lots of multiple facts derived from findings and/or conclusions from other papers. This makes the presented submission rather difficult to read. Again, a critical appraisal of the findings and conclusions from other papers would have been strongly desirable and helpful, particularly as the authors' aim is to evaluate the scientific evidence on the topic and make recommendations regarding a clinical protocol.
(4) In light of the scientific evidence on the topic of indirect restoration repair not having been properly appraised the resulting clinical protocol presented by the authors is only of extremely limited value.

Experimental design

In view of number of databases available for doing scientific reviews in dentistry it is unclear why the authors have limited themselves only to one database, PubMed. The omission of other essential databases (e.g., Embase, Web of Science, Scopus, and the Cochrane Library databases) is particularly unfortunate and regrettable because the authors' stated aim is to investigate the scientific evidence of the topic in question. Limiting themselves to only one search database does NOT allow for investigation of the scientific evidence that is available on the topic. In addition, hand search in the nonpeer-reviewed literature ProQuest and ClinicalTrials.gov. would have been advisable to meet the authors' aim of investigating the scientific evidence on the topic in question.

As the authors have identified 38 papers on different resin composite intraoral repair protocols that included mechanical and chemical aspects, for resin based, glass-ceramic or zirconia restorations, it is unclear why this review was not done using a systematic approach. Only a systematic review, ideally with meta-analysis, would have achieved the aim of the authors - investigating the scientific evidence.

Validity of the findings

In view of the fundamental shortcomings and flaws mentioned above the validity of findings of this review is very low. In fact, its very limited validity does not warrant the clinical protocol recommendation made by the authors.

Additional comments

In view of the above comments, this manuscript does not meet the aims and objectives of the journal. As a result, in the opinion of this reviewer, publication cannot be recommended.

·

Basic reporting

1) The first paragraph of the introduction starts well by mentioning ceramic and resin materials as the first choice for CAD-CAM. However, here are some observations regarding its content:
a) Is the statement about being the first choice only applicable in chairside procedures (line 79)? Why mention this? Wouldn't it be equally applicable to labside procedures? Is labside not even more common than chairside? Please consider if this statement really necessary.
b) The second sentence presents characteristics of CAD-CAM using comparative terms, stating that it "STANDS OUT" (line 80), is "LESS time-consuming," and "REDUCES manufacturing variability." To which specific technique does this comparison refer? Alternatively, consider presenting the idea in a non-comparative manner using explanatory expressions.

2) About the use of the terms "TZP", "TPZ" and "PSZ" - In the survey methodology, the authors use the terms "TZP" and "PSZ" as keywords to refer to dental zirconias. However, in the results section, the authors repeatedly use the term "TPZ." It is crucial to employ uniform terminology throughout the manuscript. In line 375 another expression is used (Yz’s). If different terms are used, it is advisable to provide previous explanations or justifications based on existing literature, especially considering that these terms are commonly confused and misused in the scientific community.

3) Lines 329-330: Check the writing

4) Lines 339-340: Check the writing


Despite the considerations mentioned above, the manuscript is written clearly and without ambiguities, with well written English language. The literature references are sufficient and well-explored, providing a robust background and supporting the authors' line of reasoning. Therefore, I believe that this review is crucial to support significant decision-making in dental clinical practice, aligning with the scope of this journal.

Experimental design

An observation to the survey methodology pointed to the selection of the keywords. Why did the authors added few commercial names to the survey (“lava ultimate”, “tetric cad”, “cerasmart”)? It is worth an explanation on why selecting these specific materials as keywords of the survey.

Besides that, the study design is well delimitated and logically organized. Sufficient detail and sources are adequately provided.

Validity of the findings

This is a very important subject directly related to the routine of practitioners, with considerable lack on the literature, which makes it an important subject for reviewing. If there’s any improvement I could suggest on this matter, it would be in concert about the delimitation of the objectives (lines 166-171), that could be written in a more cohesive way, maybe listed on a numbered sequence. The conclusion should be closely linked to the listed objectives.

Additional comments

The results section is well written with a good narrative. However, in the Glass-ceramic section, there are a few other points to consider:

1) The use of HF acid in repair situations seems contradictory. In lines 315-316, the authors state that the "use of non-toxic product seems like an essential consideration to be taken." However, in lines 283-284, the authors note that HF acid is "known for its high toxicity and even forbidden by dentists to have it on clinics in many countries." Additionally, in lines 290-291, it is said that "This is the most explored repair method in the literature, and tends to be the most successful method when bonding glass-ceramic materials to direct resin". Reading the second paragraph from the beginning, it is comprehensible that the extraoral use of HF acid is acceptable, and that the intraoral use of it is harmful to soft tissues. Then it is linked directly to direct repair with silane and stating that "this is the most explored repair method in the literature." This part doesn’t seem very consistent. A link is lacking to improve the comprehension of the writing. A suggestion would be to end the initial speech on conventional adhesion with extra-oral surface treatments, and then start again by saying that it can also be done intra-orally, but that care must be taken to carry out the procedure in absolute isolation. Perhaps just changing the order of the text could be more comprehensive.

2) Lines 282-283: The authors confirm that the main etchant used is 15-2.5% HF acid; however, the most used concentrations usually goes around 5-10%. The mentioned reference (Tian et al, 2014) assesses studies that evaluated the use of HF in different concentrations, but does not say about being the most used ones. The authors could be more careful about this statement.

---

## Round 0.2 · accepted · Accept

Congratulations and all the best.

·

Basic reporting

no comment

Experimental design

no comment

Validity of the findings

no comment

Additional comments

The authors made all the suggested changes. The changes made the manuscript easily to be read. It seems suitable for publication now.